# Optimized Design of Irrigation Water-Heating System and Its Effect on Lettuce Cultivation in a Chinese Solar Greenhouse

**DOI:** 10.3390/plants13050718

**Published:** 2024-03-04

**Authors:** Liangjie Guo, Xinyi Chen, Shiye Yang, Ruimin Zhou, Shenyan Liu, Yanfei Cao

**Affiliations:** 1College of Horticulture, Northwest A & F University, Yangling 712100, China; glj18503443835@163.com (L.G.); chenxyiii@nwafu.edu.cn (X.C.); ysy0819@nwafu.edu.cn (S.Y.); zhouruimin@nwafu.edu.cn (R.Z.); liuas@nwafu.edu.cn (S.L.); 2Key Laboratory of Protected Horticultural Engineering in Northwest, Ministry of Agriculture and Rural Affairs, Northwest A&F University, Yangling 712100, China

**Keywords:** irrigation water-heating system, Chinese solar greenhouse, warm-water irrigation, lettuce

## Abstract

In cold regions, the low irrigation water temperature is an important factor of low-temperature stress for greenhouse crops. In this paper, an irrigation water-heating system (IWHS) is proposed to increase the water temperature by utilizing the excess heat in the solar greenhouse. The heat-collection capacity of the system was analyzed by screening the IWHS process parameters in a Chinese solar greenhouse, and a warm-water irrigation experiment for lettuce was conducted. The results demonstrated that the water temperature increased with the increase in wind speed, and the increase in daily average water temperature reached the maximum value of 8.6 °C at 4.5 m/s wind speed. When the heat exchanger was installed at a height of 3.0 m, the collector capacity increased by 17.8% and 6.0% compared with the heating capacity at 0 m and 1.5 m, respectively, and the operation termination water temperature was 22.0–32.2 °C and its coefficient of performance (COP) was optimal. Surface darkening of the heat exchanger did not affect the heat-collection capacity of the system. Using the IWHS effectively improved the temperature of lettuce irrigation water in the Chinese solar greenhouse. The increased frequency of warm-water irrigation significantly promoted lettuce growth and increased the average yield per plant by 15.9%. Therefore, IWHS effectively increased the irrigation water temperature in a Chinese solar greenhouse in winter. Improving the system would enhance its economic and application value.

## 1. Introduction

Protected horticulture has developed rapidly in China, the Netherlands, Israel, the United States, Spain, and other countries over the past 60 years, becoming an important national economic industry [1]. By 2022, the total horticultural area in China was >2.8 million ha, accounting for >80% of the total protected horticulture area globally. Chinese solar greenhouses accounted for 29% of the total Chinese horticultural area, effectively ensuring an annual stable supply of agricultural products, such as vegetables, fruits, and flowers, in northern China [2]. Areas of northern China that feature a cold climate result in winter irrigation water being colder than the minimum irrigation temperature required for plant growth [3]. Furthermore, practical irrigation water-warming measures in actual production are lacking. Therefore, improving irrigation water-heating technology to solve the issue of low-temperature stress of Chinese solar greenhouse crops and improve greenhouse production efficiency is of great significance to horticultural production in cold areas.

Temperature is the microclimate factor that affects crop growth the most in Chinese solar greenhouses [4], and root zone temperature has a greater effect on plant stress than air temperature [5]. Numerous studies have been conducted on root zone temperature regulation methods and strategies [6,7,8,9,10,11]. Furthermore, some studies have analyzed the effects of root zone temperature on crop physiology and ecology [12,13,14,15], nutrient absorption [16], and yield [15]. Some studies confirmed that root zone temperature regulation positively affected crop growth [17,18,19], and the irrigation water temperature was one of the main factors affecting root zone temperature. Several researchers mainly studied the influence of warm-water irrigation on flowers and field crops [20,21], and confirmed the significance of warm-water irrigation for crop growth. For vegetable facilities, warm-water irrigation research was mainly conducted on leaf vegetable crops [22]. Hooks et al. conducted a hydroponic experiment on lettuce in winter [23], and reported that a nutrient solution heating temperature of 22 °C increased the yield by 31.4% compared with no heating, indicating that increasing the winter root zone temperature positively affected lettuce growth. The appropriate water supply is the crucial factor for obtaining a high yield and quality of vegetable crops [24,25], as water regulates the physiological and biochemical state of the plant under both normal and stressful conditions [26].

The irrigation water-heating measures widely used in production mainly include solar water pools [27] and burning non-renewable resources [28]. Although the above heating methods have many applications, they are subject to limitations. For example, building a solar water pool in a solar greenhouse requires a certain amount of greenhouse land. Additionally, the tight coverage of the pool raises the water temperature slowly, and the increase is small, which cannot meet irrigation requirements. The coverage is not exact, and increases the greenhouse air humidity, which easily induces high-humidity diseases. Some heating devices heat irrigation water through electric heating or burning non-renewable resources. Although the heating capacity is strong and the heating speed is fast, the cost is high, which is not conducive to energy saving and environmental protection. Furthermore, the irrigation water-heating system (IWHS) based on solar photovoltaic/thermal technology [29] has good overall performance but high investment and maintenance costs.

The greenhouse air temperature is relatively high at approximately noon in winter, and contains abundant air heat energy [30]. Many researchers have used water as a medium to collect air excess heat and solar energy and have designed various greenhouse heat-collection devices [31,32]. The air excess heat heating technology of solar greenhouses includes the Earth–gas heat exchange system [33] and air waste heat pump technology [34], which can store excess greenhouse heat during the day through “peak cutting and valley filling” and is used to increase the root zone temperature and air temperature. Based on the above research, this experiment used greenhouse air excess heat and insulation technology to improve the solar greenhouse irrigation water temperature in winter and spring to develop a suitable IWHS.

Therefore, this study proposed an IWHS. This experiment screened the wind speed, height, and other process parameters of the heat exchange device, and analyzed the system application effect on lettuce to determine the best IWHS process parameters and explore the effects of the IWHS on lettuce growth. It is hoped that a new method and practical reference for warm-water irrigation of solar greenhouse lettuce in winter will be established.

## 2. Results and Analysis

### 2.1. Effects of Different Wind Speeds on the Heating Effect of the System

Wind speed is an important factor affecting heat transfer, and the appropriate wind speed determines the heat-collection performance of the system. This experiment used three wind speeds in the screening test. The heat-collection capacity of the system increased with the increase in wind speed when the temperature was higher on five sunny days, and the run time was 3.0 h. At wind speeds of 2.3, 3.4, and 4.5 m/s, the average daily increase was 7.7, 8.3, and 8.6 °C, respectively (Table 1). Compared with the wind speeds of 2.3 and 3.4 m/s, the 4.5 m/s wind speed increased the water temperature by 11.7% and 3.6%, respectively. The faster wind speed increased the system ventilation volume, promoted the rapid exchange of more heat, and improved the system heating capacity.

### 2.2. Effects of Heat Exchangers of Different Heights on the Heating Effect of the System

Due to the influence of the solar greenhouse back roof and insulation quilt, the distribution of solar radiation and air temperature at different heights near the back wall of the greenhouse varied. Therefore, it was important to screen the heat exchanger height. When the system ran for 4.5~6.0 h in the maximum daily temperature period, the heat-collection performance of the system was T_h3_ > T_h2_ > T_h1_. The T_h1_, T_h2_, and T_h3_ water temperatures increased by an average of 9.0, 10.0, and 10.6 °C, respectively (Table 2). The T_h3_ water temperature increased by 17.8% and 6.0%, respectively, compared with that of Th1 and T_h2_, and the T_h3_ water temperature at the end of the operation could reach 22.0~32.2 °C. On 31 October 2022, reducing greenhouse ventilation achieved a maximum water temperature of 32.2 °C. Thus, the system heat-collection efficiency was improved by reducing ventilation without affecting normal crop growth. However, the T_h1_ total solar radiation was 2.4 and 1.2 times that of T_h3_ and T_h2_, respectively, and the system heat-collection ratio was opposite to that of the heat system collection, indicating that solar radiation did not significantly affect the system heat collection. The air inlet temperature at different heights was monitored on 19 November. The average inlet air temperatures of T_h1_, T_h2_, and T_h3_ were 22.2, 22.9, and 23.4 °C, respectively, which positively correlated with the temperature rise. Therefore, the heat exchanger placement height should be determined according to the air temperature distribution and plant height and should be placed in the middle and upper part of the ridge height direction.

### 2.3. Effects of Heat Exchanger Surface Darkening on the System Heating Effect

Figure 1 depicts the system water temperature change after the heat exchanger surface was darkened. There was a small difference (0.1 °C) in the water temperatures between the darkened and undarkened surfaces. This analysis revealed that the low air temperature at the fan outlet resulted in a low device surface temperature, and the water did not effectively absorb the solar radiation.

### 2.4. Changes in IWHS Water Temperature in Lettuce Greenhouses

The water temperature changes of the IWHS in the lettuce greenhouse during four consecutive sunny days (28 January to 31 January 2023) were selected for analysis. Figure 2 demonstrates that the minimum and maximum greenhouse indoor air temperatures during this period were 3.8 °C and 29.0 °C, respectively, and there was a period when the indoor temperature was lower than the minimum growth temperature requirement of lettuce. In the morning, the indoor air temperature rises. The IWHS water temperature gradually rises when the IWHS meets the control strategy and begins operating. The water temperature increase rate slows when the difference between the air and water temperatures decreases. The average IWHS maximum water temperature for four days was 23.2 °C, reaching the suitable water temperature for lettuce irrigation, which is 11.2 °C higher than that of unheated water. Due to normal greenhouse ventilation, the indoor temperature was above 26.0 °C for a short time at noon, and IWHS heating capacity was limited. The indoor temperature decreased as the solar radiation intensity decreased, and the system stopped operating. When the IWHS stopped, the water temperature decreased by 2.8~3.1 °C per day, indicating that the thermal insulation tank capacity was limited. The thermal insulation capacity can be improved by increasing the thermal insulation cotton thickness.

### 2.5. Heat-Collection Performance of IWHS in Lettuce Greenhouse

The IWHS heat-collection capacity and overall energy consumption were calculated using the test data of four sunny days during the test period (Table 3). The system heat-collection capacity was 5.60~10.90 MJ and the average COP was 3.07. On 19 January 2023, the initial water temperature was higher and the system heat collection was lower due to the higher air temperature that day resulting in a higher COP than that in other periods.

### 2.6. Effects of Warm-Water Irrigation on Lettuce Soil Temperature

Figure 3 depicts the effects of warm-water irrigation on the lettuce soil temperature. The soil temperature data during the irrigation period from 19 January to 20 January 2023 were analyzed. Figure 3 demonstrates that warm-water irrigation increased the lettuce soil temperature by 4.0 °C at most, and the average soil temperature of the experimental group was 1.2 °C higher than that of the control group within 6.0 h of irrigation treatment. Subsequently, the influence of irrigation water temperature on soil temperature gradually decreased as the irrigation treatment time increased, and the temperature difference between the experimental and control groups gradually became consistent.

### 2.7. Effects of Warm-Water Irrigation on Lettuce Growth Indexes

Figure 4 depicts the changes in lettuce plant height and stem diameter after warm-water irrigation. The control area (CK) and test area (T) plant heights exhibited significant differences after 35 days of warm-water irrigation. At harvest, the CK and T plant heights were 20.33 cm and 23.26 cm, respectively, increasing by ~9.6%. The CK and T stem diameters exhibited significant differences after 42 days of warm-water irrigation. At harvest, the CK and T stem diameters were 17.07 mm and 17.89 mm, respectively, an increase of ~4.8%. The results indicated that warm-water irrigation significantly promoted the growth of lettuce plant height and stem diameter and weakened the stress effect of low temperature on lettuce growth.

### 2.8. Effects of Warm-Water Irrigation on Lettuce Root and Leaf Indexes

Table 4 reports the effects of warm-water irrigation on the lettuce root index. The warm-water irrigation significantly increased the lettuce root surface area, the number of root tips, the number of leaves, and the leaf area, and T increased by 16.4%, 14.3%, 8.7%, and 16.7%, respectively, compared with CK. The results demonstrated that warm-water treatment significantly promoted lettuce root and leaf growth, and improved nutrient and water absorption.

### 2.9. Effects of Warm-WaterIrrigation on Lettuce Yield

Table 5 reports the effects of warm-water irrigation on lettuce yield. Warm-water irrigation significantly increased the yield per lettuce plant, which was increased by 15.9% compared with the control group. The results demonstrated that warm-water treatment significantly increased the lettuce yield and farmers’ income.

## 3. Discussion

In this experiment, continuous monitoring of the water temperature of a small solar water bucket revealed that it had poor heating capacity and required a long duration, while nighttime and cloudy days decreased the water temperature rapidly to the initial water temperature. Yue et al. [35] used a large solar water bucket, and only raised the water temperature for irrigation in the entire production period of strawberries by 1~2 °C, which also indicated that the solar water bucket had a poor heating capacity. The IWHS rapidly increased the water temperature for irrigation, and the water storage barrel has a thermal insulation effect in low-temperature environments with less heat loss. The process parameter screening experiment of the system determined that the faster wind speed increased the system ventilation volume, promoted the rapid exchange of more heat, and improved the system heating capacity. However, higher wind speeds result in redundant work by the system, reducing the system coefficient of performance (COP) and adversely affecting normal crop growth [36]. Therefore, the experiment did not conduct a larger wind speed optimization test. The total solar radiation was not positively correlated with the system heat-collection efficiency, but was directly proportional to the air temperature of the fan air inlet. The analysis revealed that the heat exchanger surface temperature was low due to the cold wind blowing out after heat exchange with cold water on the outer surface of the heat exchanger, and the heat absorbed by solar radiation on the heat exchanger surface was rapidly lost without exchanging with water. Even darkening the device surface did not increase the system surface temperature to allow heat exchange with water.

Preliminary testing of the IWHS heat-collection performance revealed that the lettuce greenhouse system COP was 3.07, while Song et al. [37] reported that the average COP of the thermal collecting and releasing system developed with fan-coil units was 7.5. The main reason for this was that different amounts of water were heated, and more water could maintain a larger difference between water temperature and air temperature, resulting in higher heat-collection efficiency. In the initial heat-collection stage, the test system COP was also >10. The heat-collection efficiency decreased as the temperature difference between water and air decreased, and the COP gradually decreased, resulting in a low overall heat-collection COP. Furthermore, the maximum greenhouse air temperature during heat collection was 27.4 °C, which was lower than the 33.7 °C of the greenhouse reported by Song et al. [37]. Additionally, the water flow rate positively correlated with the system heat-collection performance. A larger water flow rate resulted in a larger heat-collection power and COP of the system. The maximum water flow rate in this test was 0.59 m/s, which was lower than that of the other side (1.2 m/s). The analysis of the application effect of warm-water irrigation on lettuce determined that the test results were consistent with previous studies that used warm-water irrigation to promote tomato and cabbage growth. These analysis findings may be because warm-water irrigation promotes the absorption of soil nutrients by plants, thereby promoting growth [16]. Furthermore, the influence of warm-water irrigation on the soil temperature of lettuce was consistent with the study of Deng et al. [38].

Previous studies reported that the optimal COP of electric hot water was ~0.9. If 100 m^3^ of irrigation water must be heated by 15 °C during the winter production period, the electric heating method requires 1933 KW·h electricity, while the IWHS only requires 567 KW·h electricity. The agricultural electricity price in Shaanxi Province is approximately 0.5 RMB/(KW·h), implying savings of 683 RMB. The system heat-collection efficiency can be improved by reducing the greenhouse ventilation duration to reduce heat loss and improve the device performance without affecting normal crop growth. Due to the relatively low irrigation frequency in winter, subdivision irrigation can also reduce equipment input. The system fan has an active air exchange function, and the heat exchanger has a heat release function. Subsequent research can integrate the heat collection, heat release, and active dehumidification functions of the fan to improve its functionality and economy.

Although a suitable irrigation water temperature benefits plant growth, a new device application necessitates systematic research, establishing relevant basic databases for different plants, growth stages, and environmental temperatures, and studying the most energy-efficient operation strategy and most efficient irrigation measures to improve the system application effect. Follow-up research should examine the temperature field and water spatial distribution characteristics of the root zone under warm-water irrigation for different cultivation modes, such as soil, substrate, and coconut bran cultivation, and establish the corresponding mathematical models. In addition, we also need to study optimal control strategies for root zone temperature and air temperature to provide a more suitable growth environment for plants, reduce the impact of low temperature stress on plant growth, and promote increased yields and income [39].

## 4. Materials and Methods

### 4.1. The IWHS

The IWHS comprises a heat exchanger, a control electric box, water circulation pipes, a water pump, and a heat preservation storage bucket (Figure 5). The water in the heat exchanger flows from the bottom mouth to the upper mouth. In the high temperature period in the solar greenhouse, the fan and circulating water pump operate simultaneously under the condition of meeting the control strategy. As the air temperature is higher than the water temperature, the heat stored in the air is transferred to the water by forced convection heat transfer in the heat exchanger to heat the irrigation water in the thermal storage bucket.

### 4.2. IWHS Process Parameter Screening

#### 4.2.1. Experimental Greenhouse

The IWHS process parameter screening was conducted from 31 October to 21 December 2022, in the horticulture field of Northwest A&F University, Yangling District, Shaanxi Province (34°17′ N, 108°05′ E). The Chinese solar greenhouse spanned 10.5 m, the east–west length was 15.0 m, and the ridge height was 5.8 m. The rear gravel wall was 3.6 m high.

#### 4.2.2. Experimental Method

Monitoring of Water Temperature in Solar Water Buckets in the Chinese Solar Greenhouse

The experiment placed a polypropylene polymer (PP) bucket in the middle of the ground of the greenhouse. The bucket volume was 120 L and the bucket contained 100 L of water. The water temperature in the middle of the bucket was determined using a PT100 temperature sensor at 10 min collection intervals. The test monitored the water temperature change in the small solar water bucket (Figure 6). At a maximum indoor air temperature of 26.6 °C and minimum air temperature of 4.4 °C, the water temperature in the solar water bucket increased from the highest 11.6 °C on the first day to 14.7 °C for five consecutive days, and the average water temperature increased by 1.9 °C during the day and decreased by 1.2 °C at night. When the water temperature decreased to 11.7 °C after a cloudy day, it was only 0.1 °C higher than the highest water temperature on the first day, and the increase rate was slow; this was easily affected by low-temperature weather, and it was difficult to reach the appropriate irrigation water temperature. Simultaneously, the small amount of test water resulted in a faster heating rate, while the heating rate in the large solar water bucket was slower, and it was more difficult to meet the required irrigation requirements.

Screening of IWHS Fan Speed

The fan wind speed screening test used three sets of systems to set the wind speeds of 2.3, 3.4, and 4.5 m/s (two repetitions, five consecutive measurements at each time point, and the average value was taken), and other conditions were consistent. The heat exchanger was a harmonica-type, and the overall size was 500 mm × 600 mm × 50 mm. The polypropylene random (PPR) water circulation pipe had a diameter of 32 mm. The pump flow rate was 27 ± 1 L/min. The thermal insulation storage buckets were covered with 10 mm thick thermal insulation cotton with aluminum foil on the exterior.

Screening of IWHS Heat Exchange Device Height

The air temperature and solar radiation intensity vary at different times and spatial locations within the greenhouse. Therefore, process parameter screening for different heat exchanger heights was conducted. The heat exchanger was positioned at the midpoint of the greenhouse length, 1.5 m from the back wall. The heights were set at 0, 1.5, and 3.0 m above the ground. The water flow rates for all three heights were maintained at 27 ± 1 L/min and with a wind speed of 4.5 m/s. Figure 7 depicts the side view of the experimental measurement points.

Screening of IWHS Heat Exchanger Surface Color

The surface color screening test of the heat exchanger involved darkening the outer surface of the device with black paint (experimental group), and the control group was the undarkened surface. The other conditions were the same; the distance between the system and the north wall was 1.2 m, and the influence of the system position was ignored to examine the influence of the device surface darkening on the temperature-increase effect of the system.

#### 4.2.3. Relevant Sensor Testing

The PT100 platinum thermal resistance temperature sensor was used to measure the test greenhouse air temperature (1.5 m above the ground), the water temperature in the middle of each water bucket, and the fan inlet air temperature. The PT100 platinum thermal resistance temperature sensor data were collected using a TCP-518D temperature acquisition module (Gekong Electronics, Beijing, China). The total solar radiation on the heat exchanger surface was measured using the total solar radiation sensor. Water flow was measured using an electronic fuel meter, while the fan operating wind speed was measured using a smart anemometer (Table 6).

### 4.3. Application Effect Test of IWHS in Lettuce Cultivation

#### 4.3.1. Experimental Greenhouse

The IWHS was used to study the application of warm-water irrigation on lettuce in the Chinese solar greenhouse of Modern Agriculture Innovation Park in Yangling District, Shaanxi Province (34°30′ N, 108°03′ E) from 7 December 2022 to 7 February 2023. The greenhouse runs east–west, with a length of 48 m, clear span of 9.6 m, ridge height of 5.0 m, sinking depth of 0.4 m, and rear wall height of 4.0 m. The wall is made of clay brick masonry, and the front greenhouse roof is covered by thermal insulation at night. During the test, natural ventilation was achieved using bottom- and top-rolled film ventilation windows. The insulation was lifted at around 08:30 in the morning and dropped at around 5:00 pm. Figure 8 depicts the section diagram of the experimental greenhouse.

#### 4.3.2. Experimental Method

In the system application effect experiment, the test material was loose-leaf lettuce of the variety GRAND RAPIDS TBR. Three-leaf, one-core seedlings of robust growth and uniform size were selected for planting on 7 December 2022. A single cell area was 12.75 m^2^ (1.5 m × 8.5 m), and the plants were spaced 30 cm × 30 cm apart (Figure 9). Three cells were established in each T and CK area, and one cell was established at the junction as the quarantine area. Soil cultivation and drip irrigation were used; the irrigation strategy was artificial irrigation under sunny conditions according to weather conditions. The water temperature in the experimental group was ≥23 °C, and the management measures were the same except for the different irrigation water temperatures. According to Li et al. [40], the optimal lettuce root zone temperature is 22~25 °C. The system operation mode was set as follows: the system operated when the air and water temperature controllers detected that the air temperature and the water temperature of the water storage bucket were >26 °C and <23 °C, respectively. The water flow and hot air circulated for rapid heat exchange, and the water temperature continued to rise. The system ceased operating when the temperature controllers detected that the water temperature was >24 °C or the air temperature was <25 °C.

#### 4.3.3. Relevant Sensor Testing

The PT100 platinum thermal resistance temperature sensor was used to measure the water temperature in the middle of the water body of the bucket, the soil temperature of test area T and control area CK (10 cm depth), and the air temperature of the test greenhouse (1.5 m from the ground). The water flow was measured using the AWT electronic fuel meter, and the fan operating wind speed was measured using the smart anemometer.

The plant height (the distance from the base of the stem to the highest point), and the maximum leaf length and leaf width, of the lettuce were measured with a tape measure, the stem diameter was measured with a vernier caliper, and the number of leaves on each plant was recorded. During the harvest, five whole plants were randomly selected for each treatment, the water from the leaf surface and root system was dried, and the yield of each plant was determined using an electronic balance. The roots from the different treatments were rinsed with deionized water, the roots were scanned using a color image scanner, and the images were stored. The root growth index was analyzed using root analysis software (Table 7).

### 4.4. Relevant Correlation Formula

The system heat collection $Q_{C}$ was mainly expressed by the heat stored in the water storage tank and calculated as follows:(1)$$Q_{C}=\rho_{w}C_{w}V\Delta t$$
where $\rho_{w}$ is the water density in the storage bucket, 1000 kg/m^3^; $C_{w}$ is the specific heat capacity of water, 4183 J/(kg·°C); and *V* is the water tank volume (m^3^). The water quantity in the performance optimization test was 400 L, and the water quantity in the bucket in the lettuce test was 300 L. Δ*t* indicates the change in the initial and end water temperatures of the IWHS.

$Q_{e}\;$ represented the overall system energy consumption and was determined as follows:(2)$$Q_{e}={(P}_{wp}+P_{DF})\Delta t$$
where $P_{wp}$ is the power of the water pump (W), $P_{DF}$ is the fan power (W), and Δt is the system operating time.

$Q_{s}$ represented the total amount of solar radiation received by the heat exchanger surface (W/$m^{2}$) and was calculated as follows:(3)$$Q_{s}=\Delta tAI_{i}$$
where $I_{i}$ is the instantaneous solar radiation received by the plate surface at time *i* (W/$m^{2}$); and *A* is the total area of the heat exchanger ($m^{2}$), 0.3.

*COP* is commonly used to evaluate the energy consumption of a heat-collecting system and was derived as follows:(4)$$COP=\frac{Qc}{Qe}$$

η is the heat-collection ratio, which indicates the ratio of the heat collected by the heat exchanger (the heat stored in the water storage tank) to the solar radiation energy incident on its surface and was calculated as follows:(5)$$\eta =\frac{Qc}{Qs}$$

## 5. Conclusions

In this study, the irrigation water-heating system (IWHS) process parameters were designed and screened, and the system application effect on lettuce cultivation was analyzed. The results demonstrated that the average daily water temperature increased by 7.7, 8.3, and 8.6 °C under wind speeds of 2.3, 3.4, and 4.5 m/s, respectively. The system heating capacity was highest at a wind speed of 4.5 m/s. When the heat exchanger was 3.0 m above the ground, the heating capacity increased by 17.8% and 6.0% compared with the change at 0 and 1.5 m, respectively, the operation termination water temperature was 22.0~32.2 °C, and the heating capacity and COP were optimal. Darkening the heat exchanger surface did not affect the system heat-collection capacity. Using IWHS effectively improved the lettuce irrigation water temperature. The system COP was 3.07 in the lettuce greenhouse, and the increased frequency of warm-water irrigation significantly promoted lettuce growth and increased the yield per plant by 15.9%. Further optimization and system improvement are expected to be applied in practical production.

## Figures and Tables

**Figure 1 plants-13-00718-f001:**
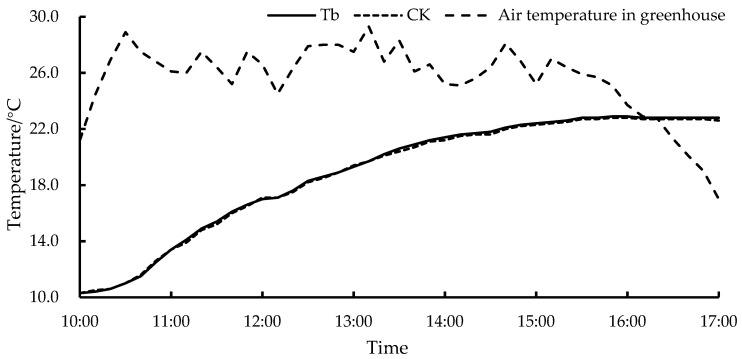
The system water temperature change after the heat exchanger surface was darkened. Tb, the system with darkened heat exchanger; CK, the system with undarkened heat exchanger.

**Figure 2 plants-13-00718-f002:**
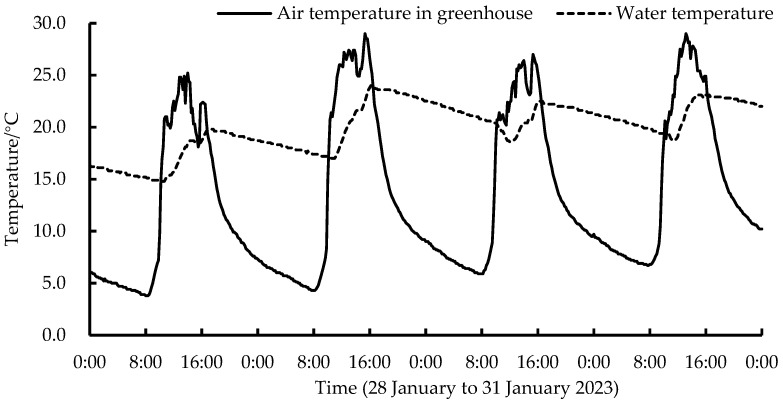
Changes in IWHS water temperature in lettuce greenhouses under sunny conditions.

**Figure 3 plants-13-00718-f003:**
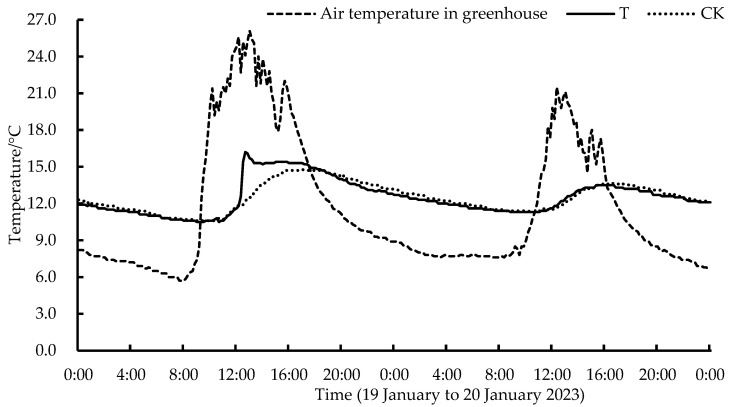
Effects of warm-water irrigation on lettuce soil temperature. T, the soil temperature in warm-water irrigation area; CK, the soil temperature in normal water irrigation area.

**Figure 4 plants-13-00718-f004:**
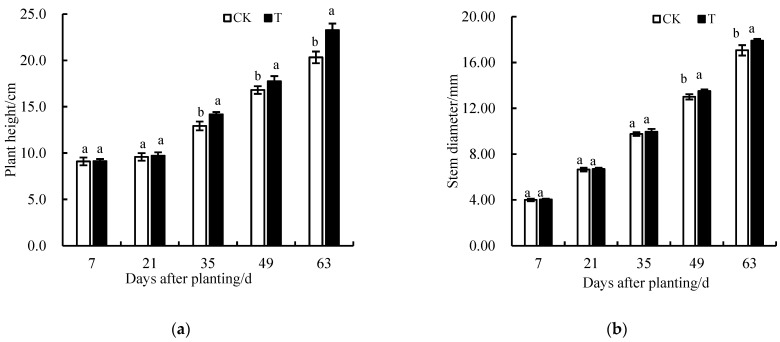
(**a**) Effects of warm-water irrigation on lettuce plant height of Lettuce. (**b**) Effects of warm-water irrigation on lettuce stem diameter. T, the experimental group plants; CK, the control group plants. Lowercase letters (a and b) indicate statistical significance by Duncan’s multiple range test (*p* < 0.05).

**Figure 5 plants-13-00718-f005:**
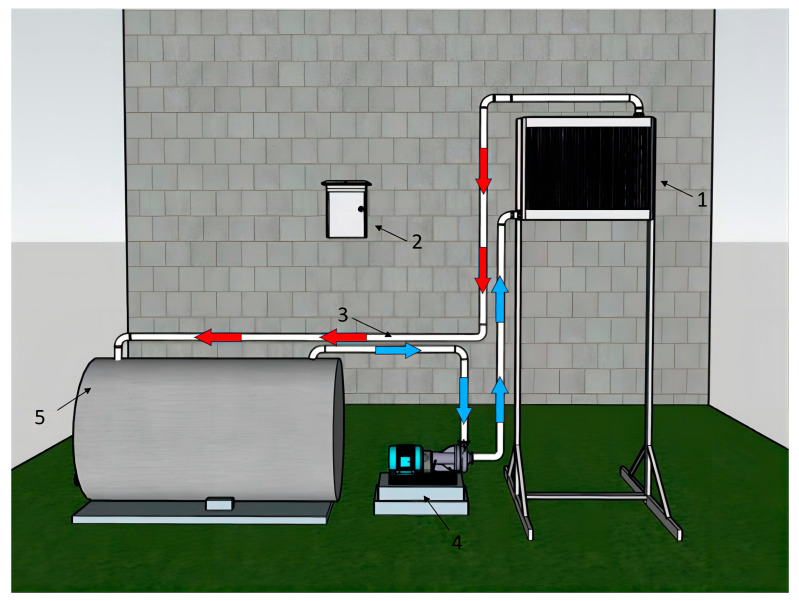
Schematic diagram of the IWHS. **1** denotes the heat exchanger comprising a fan and heat exchanger, **2** denotes the electrical control box, **3** indicates the water circulation pipe, **4** indicates the water pump, and **5** denotes the thermal insulation water storage barrel.

**Figure 6 plants-13-00718-f006:**
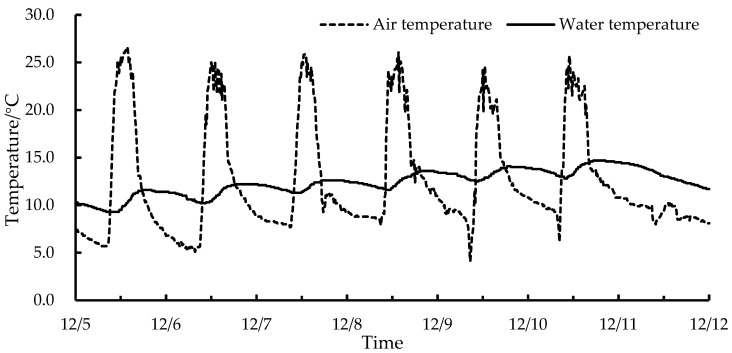
The water temperature changes in the solar water bucket in the solar greenhouse.

**Figure 7 plants-13-00718-f007:**
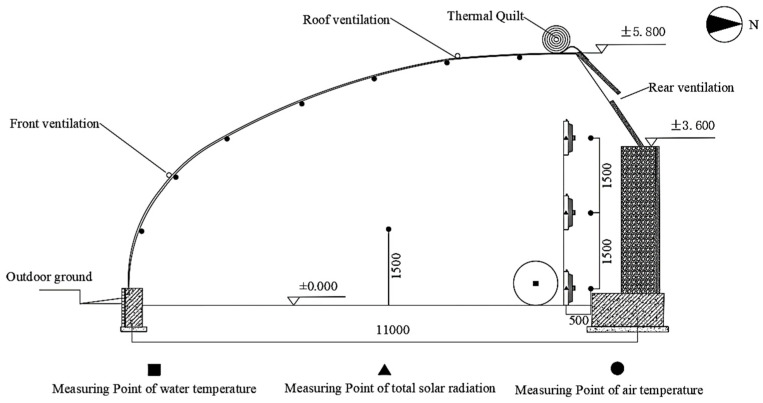
Side view of IWHS performance test measuring points at different heights.

**Figure 8 plants-13-00718-f008:**
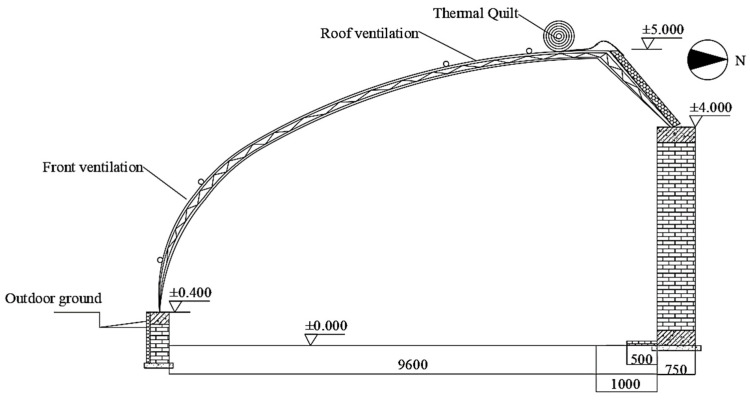
Section diagram of the experimental greenhouse.

**Figure 9 plants-13-00718-f009:**
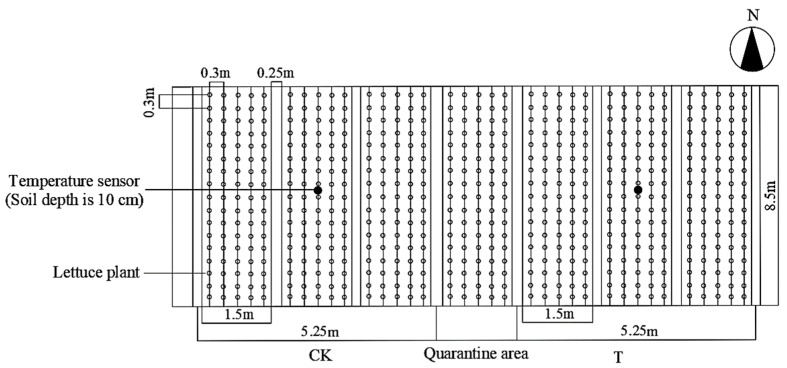
Layout plan of lettuce experimental area.

**Table 1 plants-13-00718-t001:** The effects of different wind speeds on the heating effect of IWHS.

Date	Working Time	Initial Water Temperature (°C)~ Final Water Temperature (°C)		
		2.3 m/s	3.4 m/s	4.5 m/s
5 December	11:30~14:30	9.4~17.0	9.5~17.5	9.3~17.5
8 December	10:30~13:30	10.6~17.5	10.5~17.8	10.4~18.3
12 December	11:00~14:00	11.5~20.9	11.6~21.6	11.7~22.1
13 December	10:50~13:50	12.4~18.7	12.3~19.1	12.2~19.3
14 December	10:20~13:20	10.9~19.2	10.8~20.0	10.8~20.4

**Table 2 plants-13-00718-t002:** Effect of different heights of heat exchangers on the heating effect of IWHS.

Data	Working Time	Initial Water Temperature (°C)~ Final Water Temperature (°C)			Total Amount of Solar Radiation (MJ)			η		
		Bottom T_h1_	Middle T_h2_	Top T_h3_	Bottom T_h1_	Middle T_h2_	Top T_h3_	Bottom T_h1_	Middle T_h2_	Top T_h3_
31 October	8:30~14:30	16.6~29.9	16.6~31.5	16.5~32.2	6.9	5.6	2.3	0.8	1.1	2.7
12 November	10:00~14:30	14.7~21.7	14.3~21.7	14.3~22.0	3.9	3.3	1.5	0.7	0.9	2.0
14 November	9:30~14:30	13.9~22.7	13.7~23.5	13.6~24.2	7.4	6.2	3	0.5	0.6	1.4
15 November	9:00~14:30	14.7~21.6	14.1~22.2	14.1~22.6	6.4	5.4	3.1	0.4	0.6	1.1
19 November	9:00~14:00	13.5~22.3	13.0~22.8	13.0~23.5	7.4	5.9	3.4	0.5	0.7	1.2

Note: T_h1_, the heat exchanger is 0 m from the ground; T_h2_, the heat exchanger is 1.5 m from the ground; T_h3_, the heat exchanger is 3.0 m from the ground.

**Table 3 plants-13-00718-t003:** IWHS heat-collection performance of lettuce greenhouse.

Date	Working Time	Working Hours (h)	Initial Water Temperature (°C)~Final Water Temperature (°C)	Water Temperature Rise (°C)	Heat Collection (MJ)	COP
8 December	11:00~14:30	3.50	14.4~23.1	8.7	10.90	2.06
29 December	12:42~15:12	2.50	11.8~20.3	8.5	10.60	2.96
7 January	14:10~16:20	2.17	11.9~19.2	7.3	9.10	2.79
19 January	12:50~13:40	0.83	18.0~22.5	4.5	5.60	4.47

**Table 4 plants-13-00718-t004:** Effects of warm-water irrigation on lettuce root and leaf indexes.

Treatment	Root Surface Area (cm^2^)	Root Tips (No.)	Leaves (No.)	Leaf Area (cm^2^)
CK	17.07 ± 0.90 b	757.80 ± 26.83 b	19.71 ± 0.76 b	296.34 ± 17.00 b
T	19.87 ± 1.08 a	866.00 ± 59.86 a	21.43 ± 1.13 a	345.78 ± 14.78 a

Note: T, the experimental group plants; CK, the control group plants. Lowercase letters (a and b) indicate statistical significance by Duncan’s multiple range test (*p* < 0.05).

**Table 5 plants-13-00718-t005:** Effects of warm-water irrigation on the yield of lettuce.

Treatment	Yield per Plant (g)
CK	238.76 ± 16.98 b
T	276.62 ± 9.99 a

Note: T, the experimental group plants; CK, the control group plants. Lowercase letters (a and b) indicate statistical significance by Duncan’s multiple range test (*p* < 0.05).

**Table 6 plants-13-00718-t006:** Detailed monitoring instruments information.

Measurement Metrics	Instrument	Model	Measurement Range	Accuracy	Manufacturer
Air and water temperature	PT100 platinum thermal resistance temperature sensor	WZP-GZPT-A	−50~200 °C	0.1 °C	Guizhong Technology, Guizhong, China
Total solar radiation on heat exchanger surface	Total solar radiation sensor	RS-RA-I20-AL	0~1800 W/m^2^	1 W/m^2^	Jianda Renke, Jinan, China
Circulating pipe water flow	AWT Electronic fuel meter	/	9~100 L/min	±0.5%	Xier Technology, Hangzhou, China
Fan wind speed	Smart anemometer	AS836	0.3~45 m/s	±3%	Wanchuan Electronic, Qingdao, China

**Table 7 plants-13-00718-t007:** Detailed measuring instruments information.

Instruments/Software	Model	Measurement Range	Accuracy	Manufacturer
Tape measure	GW-580E	0~5 m	0.1 mm	Great Wall Seiko, Ningbo, China
Vernier scale	/	0~150 mm	0.01 mm	Meinaite, Jinhua, China
Electronic balance	BSA224S	10 mg~220 g	0.1 mg	Sartorius, Zhangjiang, China
Color image scanner	J221	/	/	EPSON, Sakata, Japan; Regent, Vancouver, BC, Canada
Root analysis software (WinRHIZO Pro 2012a)	STD4800	/	/	
High precision electronic scale	rz-53	0.1 g~3.0 kg	0.1 g	Daming Technology, Beijing, China

## Data Availability

The raw data supporting the conclusions of this article will be made available by the authors and included in the Appendix A.

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
