# Peer review of "Optimized Design of Irrigation Water-Heating System and Its Effect on Lettuce Cultivation in a Chinese Solar Greenhouse"

_plants, 2024, doi:10.3390/plants13050718_

Round 1

Reviewer 1 Report

Comments and Suggestions for Authors

Dear authors, 

this paper is well written and important in vegetable production. It is about how to use warm water for irrigation in greenhouse for lettuce. 

It is up to date with the literature emphasing that this type of research has not be made on lettuce in general, than on flowers and filed crops. For lettuce, that is used for everyday diet it is important to enhance all growth parameters, and finally, for producers, "the yield" especially.

It is explained that wind speed, root and leaves index and yield enhance significantly using IWHS system. That is a major aim of this article and it is reached in all tested parameters.

It is good described the mechanical part of the tested system, but I have to note, Im not an expert within this field.

I recommend this article for publishing because it has all elements for improving vegetable industry of lettuce in greenhouses. 

Regards, 

Zvjezdana

Author Response

Dear reviewer:

Thank you very much for taking the time to review this manuscript.

Thank you and best regards.

Sincerely

Reviewer 2 Report

Comments and Suggestions for Authors

Interesting study. I have a few comments.

I wish this study was repeated more than one time. The data is very good, but when we look at environmental effects on crops, often 2 seasons is better to draw more robust conclusions.  However, the data seems adequate as is. I also wish sample size for weight was greater - given the large size of the study 5 plants seems small. Obviously root measurements and others cannot be done on large numbers of plants due to time limitations , but yields can be easily collected on larger numbers of plants to reduce variability. 

"Line 31,34 - Instead of "facility" horticulture, either controlled environment horticulture or protected horticulture would be better.

line 34 - hm2 would be better as ha2

Line 37 - "high-cold" is a bit awkward. Perhaps rephrase as "Areas of northern China that feature a cold climate result in winter irrigation water......"

Line 47-50 - this sentence is awkward as well- perhaps it is missing a word after "Furthermore"?

Line 56 - was was the control temperature that they authors were comparing to?  When the temp was 22oC yields increased, compared to what temp?

ling 58 replace"solar" for "sunny" as it would probably be a better English language description.

Line 283 - was the bucket color black?

Line 300 - can you add the circulation rate (from discussion) here.

Line 349 - company/source of lettuce.

Line 354 - Please add more information about irrigation - this is very important to this study - the paper states that irrigation was on/off at certain temperatures, but how long did the system operate daily? - did it just run for several hours a day as long as temperatures were adequate?  OR on cold days did it not run?  Please describe more about irrigation strategies.

Line 355 - What fertilizer program was used or if no fertilizer was used, please provide a soil test.

Figure 4 - and others - please add what test was used (Tukey, LSD, etc.) next to the P value.

Line 231 - should flow rate be m3/sec?

Line 220 and 228 - instead of "Professor Song Wei Tang's team" please rewrite to say something like Song et al. [34] reported.....

Comments on the Quality of English Language

English language could use some minor improvements as noted in above comments.

Reviewer 3 Report

Comments and Suggestions for Authors

Introduction

The author should refer to the importance of irrigation water for vegetable crops as follow

The appropriate water supply is the crucial factor for obtaining high yield and quality of vegetable crops (*), as water regulates the physiological and biochemical state of the plant both under normal or stressful conditions (**).  

*

https://doi.org/10.1007/s42729-022-00799-8

https://doi.org/10.1007/s10343-023-00947-9

**

https://doi.org/10.3390/agronomy13051227

L31: change "Facility  horticulture  has" to be Horticultural facilities have…..

L47: change 'analyzed' to be 'analyzing'

L47: change "facility vegetables" to be vegetable facilities

L54: the citation ' Panagiotis et al. ' is missing in the reference list

Results

Kindly, define the abbreviations in table footnote and figure legend

Discussion

Support this section via interpreting the findings depending on the effect of heating water in root zone on the nutrients availability   

Conclusions

Do not use the abbreviations in this section

Reviewer 4 Report

Comments and Suggestions for Authors

2. Results and Analysis should be Point 3.

3. Discussion should be point 4

4. Material and Methods should be Point 2.

100 please explain COP more.

220 and 229 citation of Prof. Song Wei tang correct in literature with year of publication.

Figures 7 and 8 are similar, one is enough to explain.

407   ….°C is missing in the text.

References should be in alphabetic order.
